# Prominin 1 and Tweety Homology 1 both induce extracellular vesicle formation

**Tristan A Bell[1,2]\*[†], Bridget E Luce[1], Pusparanee Hakim[1], Virly Y Ananda[1], Hiba Dardari[1], Tran H Nguyen[1], Arezu Monshizadeh[1], Luke H Chao[1,2]\***

[1]Department of Molecular Biology, Massachusetts General Hospital, Boston, United States; [2]Department of Genetics, Blavatnik Institute, Harvard Medical School, Boston, United States

**Abstract** Prominin 1 (Prom1) is a five-transmembrane pass integral membrane protein that associates with curved regions of the plasma membrane. Prom1 interacts with membrane cholesterol and actively remodels the plasma membrane. Membrane-bending activity is particularly evident in photoreceptors, where Prom1 loss-of-function mutations cause failure of outer segment homeostasis, leading to cone-rod retinal dystrophy (CRRD). The Tweety Homology (Ttyh) protein family has been proposed to be homologous to Prominin, but it is not known whether Ttyh proteins have an analogous membrane-bending function. Here, we characterize the membrane-bending activity of human Prom1 and Ttyh1 in native bilayer membranes. We find that Prom1 and Ttyh1 both induce formation of extracellular vesicles (EVs) in cultured mammalian cells and that the EVs produced are physically similar. Ttyh1 is more abundant in EV membranes than Prom1 and produces EVs with membranes that are more tubulated than Prom1 EVs. We further show that Prom1 interacts more stably with membrane cholesterol than Ttyh1 and that this may contribute to membrane-bending inhibition in Prom1 EVs. Intriguingly, a loss-of-function mutation in Prom1 associated with CRRD induces particularly stable cholesterol binding. These experiments provide mechanistic insight into Prominin function in CRRD and suggest that Prom and Ttyh belong to a single family of functionally related membrane-bending, EV-generating proteins.

## Editor's evaluation

This work is important because it significantly advances our understanding of membrane protein functions by establishing the mechanisms by which Prominin-1 and Tweety Homology-1 induce extracellular vesicle (EV) formation. The evidence supporting these findings is compelling as it features rigorous biochemical assays and thorough experimental validation. The study will therefore be of particular interest to researchers in cell biology and membrane dynamics, particularly to researchers with a keen interest in EV formation and its implications for cellular processes.

## Introduction

Mammalian cells interact with the extracellular environment through proteins, lipids, and glycans at the plasma membrane. Organized protrusive structures of the membrane such as microvilli and cilia are hotspots for nutrient absorption, cell cycle regulation, and extracellular signaling (*Sharkova et al., 2023*; *Satir and Christensen, 2007*). Membrane protrusions are organized both by interactions with the cytoskeleton and by sorting of proteins and lipids within the membrane bilayer (*Satir and Christensen, 2007*; *Brown and London, 2000*; *Corbeil et al., 2010*). Prominin 1 (Prom1) is a five-transmembrane pass integral membrane protein that interacts with membrane cholesterol at sites of membrane protrusion (*Corbeil et al., 2013*; *Thamm et al., 2019*). Prom1 was first characterized as the

**\*For correspondence:**
bell@molbio.mgh.harvard.edu (TAB);
chao@molbio.mgh.harvard.edu (LHC)

**Present address:** [†]Generate Biomedicines, Somerville, United States

**Competing interest:** The authors declare that no competing interests exist.

target of AC133-1, a monoclonal antibody raised against pluripotent human epithelial stem cells (*Yin et al., 1997*; *Miraglia et al., 1997*). Since then, Prom1 has been identified in the apical membranes of most epithelial and neuroepithelial cell types, but is only natively recognized by AC133-1 in stem cells and photoreceptors (*Corbeil et al., 1998*; *Kemper et al., 2010b*; *Mak et al., 2012*). In photoreceptors, Prom1 associates with a retinal cadherin (Protocadherin-21, Pcdh21) to promote normal outer segment membrane homeostasis (*Yang et al., 2008*; *Burgoyne et al., 2015*; *Rattner et al., 2001*), and several Prom1 mutations are linked to hereditary cone-rod retinal dystrophy (CRRD) (*Yang et al., 2008*; *Zacchigna et al., 2009*; *Eidinger et al., 2015*; *Liang et al., 2019*; *Maw et al., 2000*). Most animal genomes also encode at least one paralogous prominin gene (Prominin 2) that is expressed in non-retinal cell types and localizes to positively curved membrane regions (*Fargeas, 2013*).

When overexpressed, Prom1 dramatically reorganizes the plasma membrane of cultured mammalian cells into long protrusions (*Hori et al., 2019*; *Röper et al., 2000*). Small-molecule inhibitors of actin and tubulin do not impair the tubulation phenotype, indicating that cytoskeletal interactions are not strictly required for membrane bending (*Hori et al., 2019*; *Röper et al., 2000*). However, cells treated with cholesterol biosynthesis inhibitors or depletion agents do not exhibit membrane tubulation, suggesting that cholesterol plays a pivotal role in the regulation of Prom1 function (*Hori et al., 2019*; *Röper et al., 2000*).

Prom1 also induces release of small extracellular vesicles (EVs, <250 nm in diameter) that bleb from the apical plasma membrane of differentiating epithelial stem cells at organized membrane protrusions such as microvilli and cilia (*Marzesco et al., 2005*; *Wood and Rosenbaum, 2015*; *Dubreuil et al., 2007*). Small Prom1 EVs can be detected in saliva, urine, semen, neural tube fluid, and lacrimal fluid of healthy adults (*Marzesco et al., 2005*; *Hurbain et al., 2022*).

Recently, the Tweety Homology (Ttyh) protein family was hypothesized to be a distant homolog of the prominins (*Hori et al., 2019*). Ttyh proteins share prominins' five-transmembrane topology but have a more minimal extracellular domain (*Campbell et al., 2000*). Most animals have three paralogous Ttyh proteins (Ttyh1, Ttyh2, and Ttyh3) (*Matthews et al., 2007*) that all are predominantly expressed in neural tissues (*Halleran et al., 2015*; *Bae et al., 2019*; *Han et al., 2019*; *Li et al., 2021*; *Sukalskaia et al., 2021*; *Melvin et al., 2022*). Notably, overexpression of Ttyh1 at the plasma membrane in cultured cells induces plasma membrane tubulation that is strikingly similar to that observed with Prom1 (*Matthews et al., 2007*; *Jung et al., 2017*; *Stefaniuk et al., 2010*; *Wiernasz et al., 2014*).

Defining mechanisms of membrane bending by Prominin-family proteins is foundational for understanding retinal disease and stem cell development. Here, we reconstitute EV formation by Prom1 and Ttyh1 to characterize their functions in a native membrane bilayer system. To our knowledge, we present the first evidence that Ttyh1 induces formation of EVs that are functionally and physically analogous to Prom1 EVs. Ttyh1 EVs are more tubulated and protein-rich than Prom1 EVs. Prom1 forms a more stable interaction with cholesterol than Ttyh1, and when depleted of cholesterol, Prom1 EVs mimic the tubulation observed in Ttyh1 EVs. A CRRD-associated Prom1 mutant (W795R) binds cholesterol more stably than wild-type protein, suggesting that a less dynamic interaction between Prom1 and cholesterol inhibits membrane bending. These findings contribute to a path toward mechanistic understanding of Prominin-family protein function in different tissues, including the role of Prom1 in retinal pathology.

## Results
### Reconstitution and purification of Prom1 EVs

To understand how Prom1 interacts with and reshapes native membranes, we sought a method to purify Prom1 without disturbing protein–lipid interactions. Because Prom1 induces EV formation in epithelial stem cells (*Marzesco et al., 2005*), we asked whether Prom1 can also form EVs in cultured cells. C-terminally Strep-tagged Prom1 was detected in immunoblots of the conditioned media (CM) from transfected cells, but not in cells treated with a no-plasmid mock transfection (*Figure 1A, B*). To validate that the ~120 kDa Strep-reactive band we observe is indeed Prom1-Strep, we immunoblotted with the Prom1-specific antibody AC133-1 (*Kemper et al., 2010a*) and observed robust staining (*Figure 1C*). To confirm that EV production is not an artifact of transfection, we introduced Prom1-Strep into Expi293 cells by lentiviral transduction to generate a stable polyclonal overexpression cell

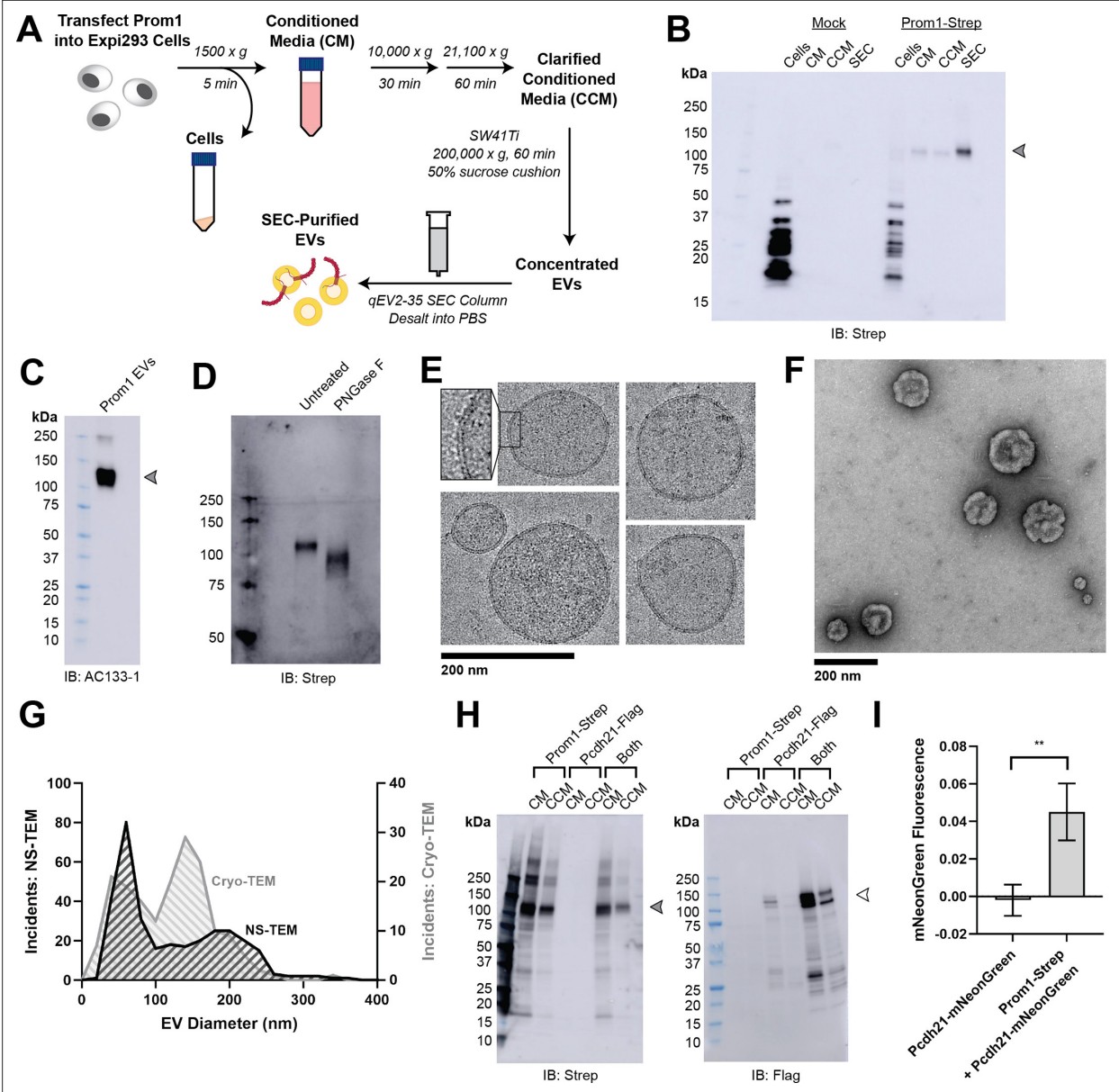

**Figure 1.** Reconstitution and purification of Prom1 EVs. (**A**) Extracellular vesicle (EV) expression and purification graphic protocol. (**B**) Anti-Strep immunoblot of cell pellet, conditioned media (CM), clarified conditioned media (CCM), or size-exclusion chromatography (SEC)-purified EVs from mock-transfected or Prom1-Strep-transfected Expi293 cells. Arrowhead indicates the expected molecular weight of Prominin 1 (Prom1). (**C**) AC133-1 immunoblot of SEC-purified Prom1 EVs. Arrowhead indicates the expected molecular weight of Prom1. (**D**) Anti-Strep immunoblot of Prom1 EVs treated with or without PNGase F to remove *N*-glycan moieties. (**E**) Cryo-transmission electron microscopy (cryo-TEM) images of purified Prom1 EVs. Inset image is magnified to emphasize membrane bilayer density. Images are lowpass filtered to 5 Å to enhance contrast. (**F**) Negative-stain transmission electron microscopy (NS-TEM) image of purified Prom1 EVs. (**G**) Measured diameters of Prom1 EVs from cryo-TEM or NS-TEM images (*n* = 322 and *n* = 176 for NS-TEM and cryo-TEM measurements, respectively). (**H**) Anti-Strep (Prom1) and anti-Flag (Pcdh21) immunoblots of CM and CCM from cells transfected with Prom1-Strep, Pcdh21-Flag, or both. Note that Pcdh21 is only detected in CCM when co-expressed with Prom1. Filled and empty arrows indicate expected molecular weights of glycosylated Prom1-Strep and Pcdh21-Flag, respectively. (**I**) Comparative fluorescence measurements of Pcdh21-mNeonGreen co-immunoprecipitated with or without Prom1-Strep (*n* = 3, **p < 0.01 by Student's two-tailed unpaired *t* test).

The online version of this article includes the following source data and figure supplement(s) for figure 1:

**Source data 1.** Raw source gel and blot images.

**Source data 2.** Labelled source gel and blot images.

**Figure supplement 1.** Prom1 induces formation of extracellular vesicles.

**Figure supplement 1—source data 1.** Raw source gel and blot images.

**Figure supplement 1—source data 2.** Labelled source gel and blot images.

line. A Strep-reactive band consistent with mature Prom1 was detectable in the CM from these cells over several rounds of cell passaging (*Figure 1—figure supplement 1A*). We assessed the protein composition of Prom1 EVs using sodium dodecyl sulfate–polyacrylamide gel electrophoresis (SDS–PAGE) and silver nitrate staining. Prom1 was a constituent of our samples alongside several other proteins (*Figure 1—figure supplement 1*). We cannot conclude whether these proteins specifically interact with Prom1 in EVs, are hitchhikers enclosed within the EVs, or originate from endogenous co-purifying EVs.

We next purified the Prom1 EVs using differential centrifugation, ultracentrifugal concentration, and size-exclusion chromatography (SEC) based on isolation methods previously described for endogenous small Prom1 EVs (*Figure 1A*; *Marzesco et al., 2005*; *Huttner et al., 1983*). It is important to note that our purification protocol removes any large EVs and midbody remnants, focusing our downstream experiments on small Prom1 EVs. Using dynamic light scattering (DLS), we measured the solution sizes of the purified Prom1 EVs. The sample was monodisperse with an average diameter of 164 ± 14 nm, considerably larger than previously reported 50–80 nm EVs measured by negative-stain transmission electron microscopy (NS-TEM) in samples purified from biological fluids (*Marzesco et al., 2005*; *Figure 1—figure supplement 1*). Upon treatment with PNGase F, an enzyme that cleaves *N*-glycan groups from proteins, the ~120 kDa Prom1 gel band shifted down to its predicted molecular weight of 102 kDa (*Figure 1D*).

To verify that the purified particles are truly EVs (secreted particles with intact bilayer membranes), we vitrified purified samples and imaged them using cryo-transmission electron microscopy (cryo-TEM) (*Figure 1E*). We observed spherical EVs with bilayer membranes (*Figure 1E*, inset). The diameters of EVs measured from cryo-TEM images were bimodal with an average diameter of 117 ± 58 nm, somewhat lower than our measurements from DLS (*Figure 1G*). We next used NS-TEM to directly compare our reconstituted Prom1 EVs with previously characterized endogenous Prom1 EVs (*Marzesco et al., 2005*; *Figure 1F*). We observed a distribution of largely circular EVs ranging in size from ~50 to ~250 nm in diameter, but more skewed toward smaller diameters than observed in the cryo-TEM data (*Figure 1G*). In addition, EVs had rough edges and internal depressions in NS-TEM, a characteristic feature of EV fixation and dehydration (*Marzesco et al., 2005*; *Figure 1G*). Because solution DLS measurements (164 ± 14 nm) suggest slightly larger EV diameters than our NS-TEM (123 ± 73 nm) or cryo-TEM (117 ± 58 nm) measurements, we speculate that sample fixation/dehydration or vitrification may induce deformation and potentially fission of the reconstituted EVs. This effect may have similarly impacted previous characterization of endogenous Prom1 EVs (*Marzesco et al., 2005*). We conclude that reconstituted Prom1 EVs have similar morphology to endogenous EVs but may be slightly larger in size.

We next asked whether Prom1 in purified EVs retains known functional behavior of Prom1 from endogenous membranes. Prom1 stably binds the retinal cadherin Pcdh21 in photoreceptors (*Yang et al., 2008*; *Burgoyne et al., 2015*; *Rattner et al., 2001*). To determine if Prom1 can co-traffic with Pcdh21 in our EV samples, we co-transfected Expi293 cells with Strep-tagged Prom1 and Flag-tagged Pcdh21 and looked for the presence of each component in cells, CM, and clarified conditioned media (CCM). Pcdh21 was only observed in CCM samples when co-expressed with Prom1, indicating that Prom1 is necessary to traffic Pcdh21 into this class of EVs (*Figure 1H*). To establish a direct interaction between Prom1 and Pcdh21, we co-expressed Prom1-Strep with mNeonGreen-tagged Pcdh21 and immunopurified Prom1 from EVs solubilized with 1% *n*-dodecyl-β-D-maltoside (DDM). Pcdh21-associated mNeonGreen reproducibly co-purified with detergent-solubilized Prom1 compared to a control condition lacking Prom1, indicating that the two proteins physically interact in purified EVs (*Figure 1I*).

## Mutations in the Prom1 transmembrane domain impair EV formation

Prom1 binds cholesterol in native membranes and the Prom1–cholesterol interaction is necessary for remodeling the plasma membrane (*Röper et al., 2000*), but the mechanism of cholesterol-dependent Prom1 function is unclear. We analyzed the sequence of human Prom1 to identify candidate cholesterol recognition amino acid consensus (CRAC, [L/V]-X$_{1-5}$-[Y/F]-X$_{1-5}$-[K/R]) or mirrored CRAC (CARC, [K/R]-X$_{1-5}$-[Y/F]-X$_{1-5}$-[L/V]) sequences in the transmembrane helices, as these motifs often predict cholesterol binding in membrane proteins (*Fantini and Barrantes, 2013*). Human Prom1 contained four CRAC and two CARC motifs, of which none were completely conserved among metazoans

and only one (CRAC-3) was modestly evolutionarily conserved (*Figure 2—figure supplement 1A*, *Figure 2—figure supplement 1*). We therefore turned to a more comprehensive evolutionary analysis of prominin proteins to identify evolutionarily conserved Prom1 transmembrane residues.

We curated prominin sequences from across eukaryotes, considering a prominin to be a sequence with five predicted transmembrane helices, two large extracellular loops, two small intracellular loops, and homology to annotated metazoan prominin sequences. Putative prominin sequences were identified across metazoans as well as in fungi, excavates, SAR (stramenopiles, alveolates, and rhizarians), and green plants (*Figure 2—figure supplement 2*, *Figure 2—figure supplement 3*). Strikingly, Trp-795 was nearly perfectly conserved across metazoa and eukaryotic outgroups, making it by far the most conserved residue (excluding Cys residues positioned to form disulfides in AlphaFold2 models) across the curated prominin sequences (*Figure 2A*, *Figure 2—figure supplement 2*). The conserved Trp-795 residue was of particular interest as a missense mutation at this site (W795R) is implicated in hereditary CRRD cases (*Boulanger-Scemama et al., 2015*).

We generated Prom1-Strep variants with mutations to disrupt each CRAC or CARC motif, the disease-associated W795R mutation, and point mutations directed against residues predicted to be in close contact with Trp-795 in Alphafold2 models (*Figure 2B*, *Figure 2—figure supplement 4*, *Jumper et al., 2021*). We then assessed EV formation by quantifying the Prom1-Strep signal secreted in EVs or retained in the cellular membranes. We found that the W795R mutation to the Prom1 TM domain significantly disrupted the proportion of Prom1 secreted in EVs (*Figure 2C*, *Figure 2—figure supplement 4* for raw blot images). This effect arose from both decreased expression of Prom1 and increased retention of protein in cells, though neither of these measurements alone met our stringent significance criteria. To better characterize the morphology of the Prom1 variants, we expressed and purified a subset of mutant Prom1 EVs at larger scale. Each of the mutants produced EVs that are 150–200 nm in diameter and monodisperse by DLS analysis, with no mutant EV sizes deviating significantly from that of WT Prom1 EVs (*Figure 2D*). Thus, transmembrane domain mutations in Prom1 primarily alter the quantity of EVs produced rather than EV size.

To better understand the mechanism of reduced Prom1 EV formation by the CRRD-associated W795R mutant, we engineered HeLa cells that stably express C-terminally StayGold-tagged WT (Prom1-SG) or W795R Prom1 (Prom1[W795R]-SG) under the EF-1α promoter. Prom1-SG signal colocalized with wheat germ agglutinin (WGA)-stained plasma membrane, but Prom1[W795R]-SG was not present at the plasma membrane and instead concentrated on WGA-positive intracellular features (*Figure 2E*). Despite the apparent low signal of plasma-membrane localized Prom1-SG, line scan analysis shows correlation between Prom1-SG signal across the cell junctions (*Figure 2—figure supplement 5*, left). In contrast, there is a marked absence of Prom1[W795R]-SG signal localized on the plasma membrane (*Figure 2—figure supplement 5*, right). We infer that mStayGold-tagged W795R Prom1 is trafficked to the plasma membrane less efficiently than the mStayGold-tagged wild-type protein.

## Prominin homolog Ttyh1 produces EVs

Ttyh proteins are proposed prominin homologs that share the five-transmembrane topology of prominins but have shorter extracellular domains (~120 amino acids in Ttyh vs ~280 amino acids in Prom) (*Hori et al., 2019*; *Figure 3A*). Prominin and Ttyh both traffic to the plasma membrane and associate with protrusive membrane structures (*Hori et al., 2019*; *Röper et al., 2000*; *Matthews et al., 2007*; *Jung et al., 2017*; *Stefaniuk et al., 2010*; *Wiernasz et al., 2014*). Given the predicted sequence homology between Prom and Ttyh, we hypothesized that Ttyh may also produce EVs. Evolutionary analysis of metazoan Ttyh proteins does not indicate conserved CRAC or CARC sites, nor does it suggest any conserved transmembrane residue analogous to Trp-795 in metazoan prominins (*Figure 3B*).

We expressed C-terminally Strep-tagged Ttyh1 in Expi293 cells and purified EVs using the same procedure as for Prom1 EVs (*Figure 1A*). We detected Ttyh1-Strep in CM, CCM, and SEC-purified EV fractions (*Figure 3C*). DLS indicated that purified Ttyh1 EVs are monodisperse with an average diameter of 180 ± 10 nm, similar to the average diameter of WT Prom1 EVs (164 ± 14 nm) (*Figure 3—figure supplement 1*).

We next characterized the morphology of purified Ttyh1 EVs using NS-TEM. We found that Ttyh EVs adopted striking long and bent tubular structures with much higher frequency than Prom1 EVs

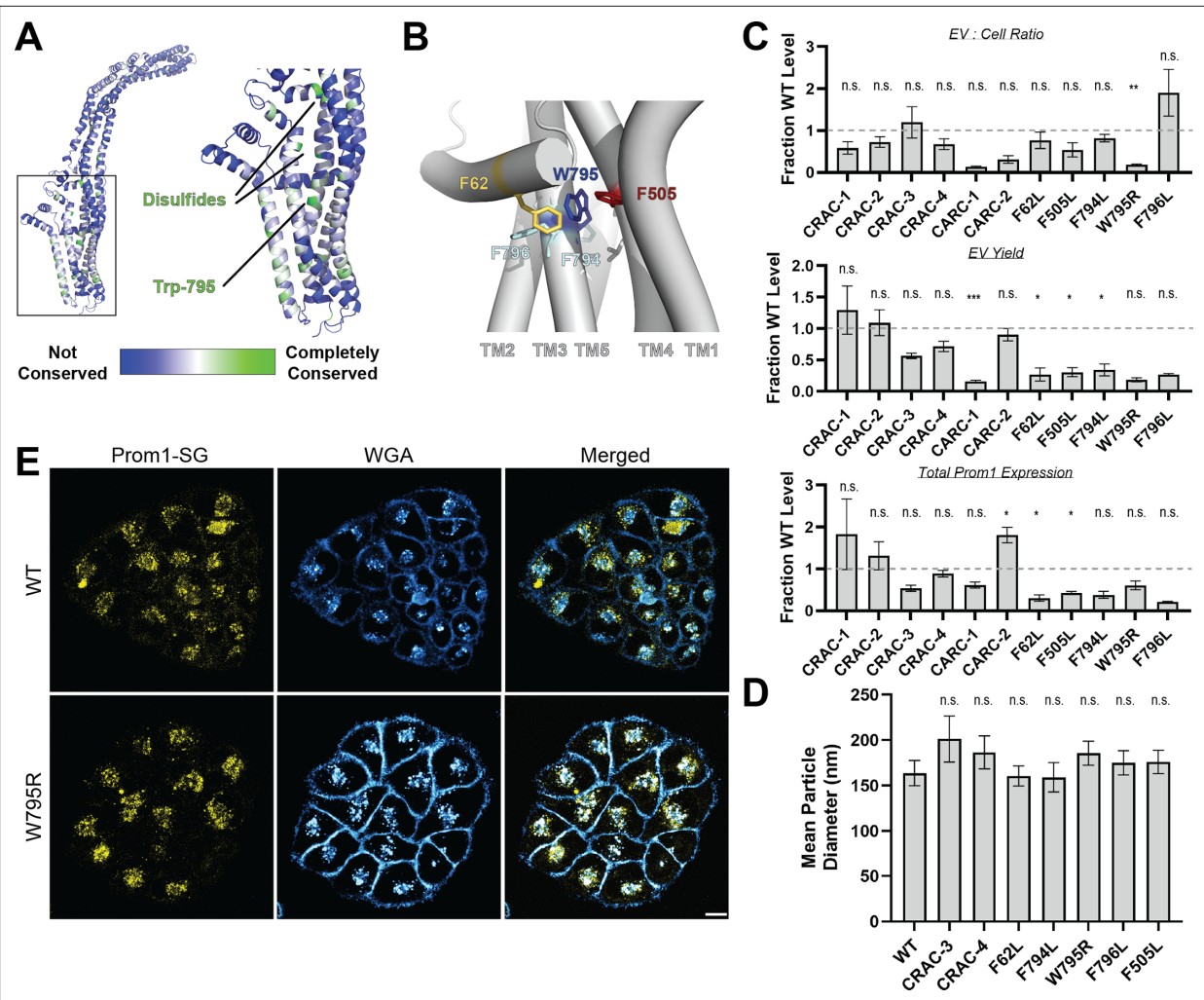

**Figure 2.** Mutations in the Prom1 transmembrane domain impair EV formation. (**A**) AlphaFold2 model of human Prominin 1 (Prom1; *Jumper et al., 2021*) with residues color coded by level of conservation across a multiple-sequence alignment of metazoan prominin sequences. (**B**) Possible network of interactions between Trp-795 and several adjacent aromatic residues in human Prom1. (**C**) Quantification of anti-Strep immunoblots of cellular and clarified conditioned media pools of Prom1-Strep mutants relative to wild-type Prom1-Strep ($n = 3$, [n.s.]$p > 0.0045$, *$p < 0.0045$, **$p < 0.0009$, ***$p < 0.00009$ by Student's two-tailed unpaired $t$ test with significance thresholds adjusted by Bonferroni correction). Raw blot images from which measurements are derived are included in *Figure 2—figure supplement 4*. (**D**) Mean particle diameter of purified Prom1-Strep EVs measured by dynamic light scattering. Error bars indicate standard deviation (SD) ($n = 5$, [n.s.]$p > 0.007$, *$p < 0.007$ by Student's two-tailed unpaired $t$ test with significance thresholds adjusted by Bonferroni correction). (**E**) Confocal fluorescence microscopy images of HeLa cells stably expressing WT (top) or W795R (bottom) Prom1-StayGold (yellow), stained with wheat germ agglutinin (WGA) (blue). Scale bar is 10 μm. Line scan traces across cell junctions are included in *Figure 2—figure supplement 5*.

The online version of this article includes the following source data and figure supplement(s) for figure 2:

**Figure supplement 1.** Putative cholesterol-binding motifs in the Prom1 transmembrane domain.

**Figure supplement 2.** Conservation analysis of metazoan Prom1 sequences.

**Figure supplement 3.** Phylogenetic analysis of Prom1 sequences.

**Figure supplement 4.** Blot images used for EV quantification in *Figure 1C*.

**Figure supplement 4—source data 1.** Raw source gel and blot images.

**Figure supplement 4—source data 2.** Labelled source gel and blot images.

**Figure supplement 5.** Line scan traces of Prom1 signal in cell lines.

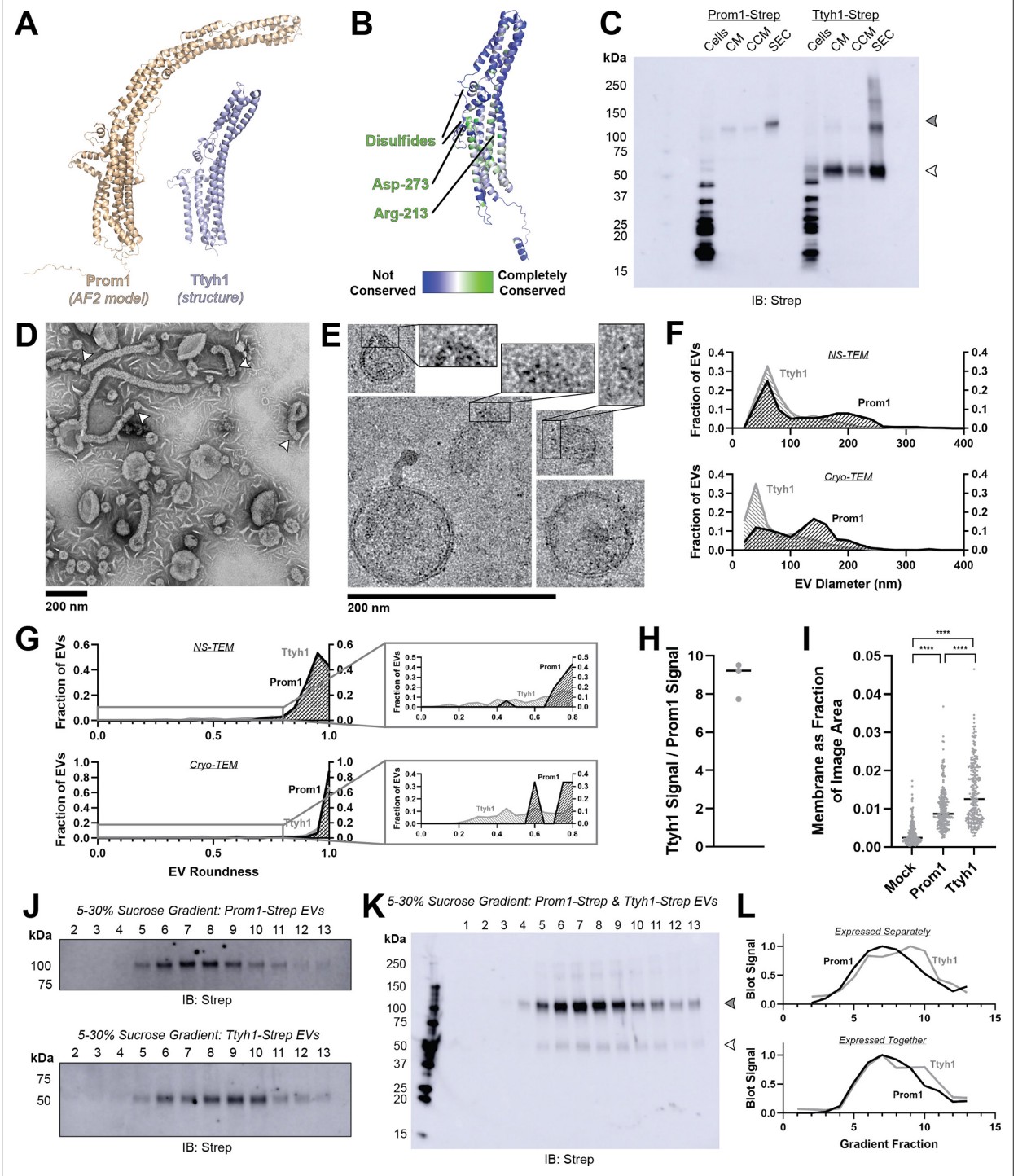

**Figure 3.** Prominin homolog Ttyh1 produces EVs. (**A**) Comparison of a Ttyh1 subunit from cryo-transmission electron microscopy (cryo-TEM) structure 7P5J (*Sukalskaia et al., 2021*) with an AlphaFold2-predicted Prominin 1 (Prom1) monomer (*Jumper et al., 2021*). (**B**) Residue-level conservation among metazoan Tweety Homology (Ttyh) proteins plotted onto a subunit of human Ttyh1. No residue analogous to Prom1 Trp-795 is present in Ttyh. (**C**) Anti-Strep immunoblot comparing Prom1- and Ttyh1-Strep extracellular vesicles (EVs) throughout different stages of purification. Filled and empty arrowheads indicate the expected positions of Prom1 and Ttyh1, respectively. Doublet and higher bands in Ttyh1 lanes are products of on-gel disulfide crosslinking in concentrated samples. (**D**) Representative negative-stain transmission electron microscopy (NS-TEM) images of Ttyh1-Strep EVs. White arrowheads indicate possible sites of EV fission. (**E**) Representative cryo-TEM images of Ttyh1-Strep EVs. Magnified insets show bilayer density at highly curved membrane segments. Images are lowpass filtered to 5 Å to enhance contrast. (**F**) Comparison of Prom1 or Ttyh1 EV diameter in NS-TEM or cryo-TEM images (*n* = 322, *n* = 1357, *n* = 176, and *n* = 2224 for Prom1 NS-TEM, Ttyh1 NS-TEM, Prom1 cryo-TEM, and Ttyh1 cryo-TEM measurements,

*Figure 3 continued on next page*

Figure 3 continued

respectively). (**G**) Comparison of Prom1 or Ttyh1 EV roundness in NS-TEM or cryo-TEM images. Secondary plots only include EVs with roundness ≤0.8 (*n* = 322, *n* = 1357, *n* = 122, and *n* = 1546 for Prom1 NS-TEM, Ttyh1 NS-TEM, Prom1 cryo-TEM, and Ttyh1 cryo-TEM measurements, respectively). (**H**) Quantification of Prom1 and Ttyh1 protein levels in EVs from anti-Strep Western blots (*n* = 3). Immunoblots used for quantification are included in *Figure 3—figure supplement 1A*. (**I**) Quantification of total EV membrane area from NS-TEM micrographs (*n* = 236, ****p < 0.0001 by unpaired Mann–Whitney test). Representative micrographs used for quantification are included in *Figure 3—figure supplement 1*. (**J**) Anti-Strep immunoblots of fractions from sucrose gradient equilibrium sedimentation of Prom1-Strep EVs (top) or Ttyh1-Strep EVs (bottom). (**K**) Anti-Strep immunoblots of fractions from sucrose gradient equilibrium sedimentation of EVs from cells co-expressing Prom1-Strep and Ttyh1-Strep. Filled and empty arrowheads indicate the expected positions of Prom1 and Ttyh1, respectively. (**L**) Quantification of immunoblots in panels K (top) and L (bottom).

The online version of this article includes the following source data and figure supplement(s) for figure 3:

**Source data 1.** Raw source gel and blot images.

**Source data 2.** Labelled source gel and blot images.

**Figure supplement 1.** Characterization of Prom1 and Ttyh1 EVs.

**Figure supplement 1—source data 1.** Raw source gel and blot images.

**Figure supplement 1—source data 2.** Labelled source gel and blot images.

(6.4% of Ttyh1 EVs vs 0.3% of Prom1 EVs, *n* = 1357 and *n* = 322, respectively) (*Figure 3D, G*). Ttyh1 EVs visualized by NS-TEM were on average smaller than Prom1 EVs (*Figure 3F, G*), and were similarly smaller than expected from solution DLS measurement (*Figure 1—figure supplement 1*). Furthermore, we observed that the smallest Ttyh1 EVs were similar in diameter to the short-axis caliper diameter of the tubular EVs (42 ± 6 nm) and we observed cases where tubular EVs appeared to be in the process of dividing into smaller EVs (*Figure 3D* white arrowheads, *Figure 3F*). Although the NS-TEM conditions deviate from a solvated physiological state, they do suggest that Ttyh1 supports comparatively greater membrane curvature than Prom1 in these EVs.

We further analyzed Ttyh1 EVs by cryo-TEM to verify that the purified sample indeed contained EVs with intact bilayer membranes (*Figure 3E*). Like Prom1 EVs, Ttyh1 EVs have a bimodal size distribution with local maxima around 60 and 140 nm (*Figure 3F*). We also observed a population of EVs exhibiting the tubular phenotype seen in NS-TEM that again substantially exceeded that seen with Prom1 EVs (8.8% of Ttyh1 EVs vs 0.8% of Prom1 EVs, *n* = 2224 and *n* = 176, respectively) (*Figure 3G*). This supports the observation that EV bending is more frequent in Ttyh1 EVs than Prom1 EVs.

Molecular crowding is known to contribute to membrane bending (*Derganc and Čopič, 2016*). To test if Ttyh1 may promote EV membrane bending through a crowding mechanism, we quantified relative protein-to-membrane ratios for Prom1 and Ttyh1 EVs. From identical amounts of transfected cell media, purified EVs contained 8.8 ± 1.0 fold more Ttyh1 protein than Prom1 (*Figure 3H*, *Figure 3—figure supplement 1*). Both Prom1 and Ttyh1 EV membrane surface areas were significantly more abundant than the background level of EVs produced in a mock-transfected control culture, but Ttyh1 produced 5.6 ± 0.8 fold more total EV membrane than Prom1 (*Figure 3I*, *Figure 3—figure supplement 1*). From these measurements, we estimate that Ttyh1 is present at ~fivefold higher concentration in EV membranes (*Figure 3—figure supplement 1*). Thus, molecular-crowding effects may be a factor contributing to the increased membrane bending we observe in Ttyh1 EVs.

To further characterize purified Prom1 and Ttyh1 EVs, we subjected purified EVs to equilibrium sucrose gradient sedimentation to resolve populations by density. We observed a single population of Prom1 EVs (centered on fraction 7) but resolved two distinct populations of Ttyh1 EVs with densities lower (centered on fraction 6) and higher (centered on fraction 9) than the Prom1 EVs (*Figure 3J*). When Prom1 and Ttyh1 were co-expressed, we observed that the resulting EVs contained more Prom1 than Ttyh1 but followed the bimodal distribution of Ttyh1 EVs (*Figure 3K*). Immunoblots of the sucrose gradient fractions showed that co-expressed Prom1 and Ttyh1 peak in the same sucrose gradient fractions, suggesting that Prom1 and Ttyh1 co-elute in the same EV populations (*Figure 3K, L*).

## Ttyh1 binds cholesterol less stably than Prom1

Because cholesterol interaction is known to regulate membrane bending by Prom1, we asked whether Prom1 and Ttyh1 bind cholesterol similarly (*Hori et al., 2019*; *Röper et al., 2000*). To comparatively measure binding, we developed a cholesterol co-immunopurification assay (hereafter referred to as

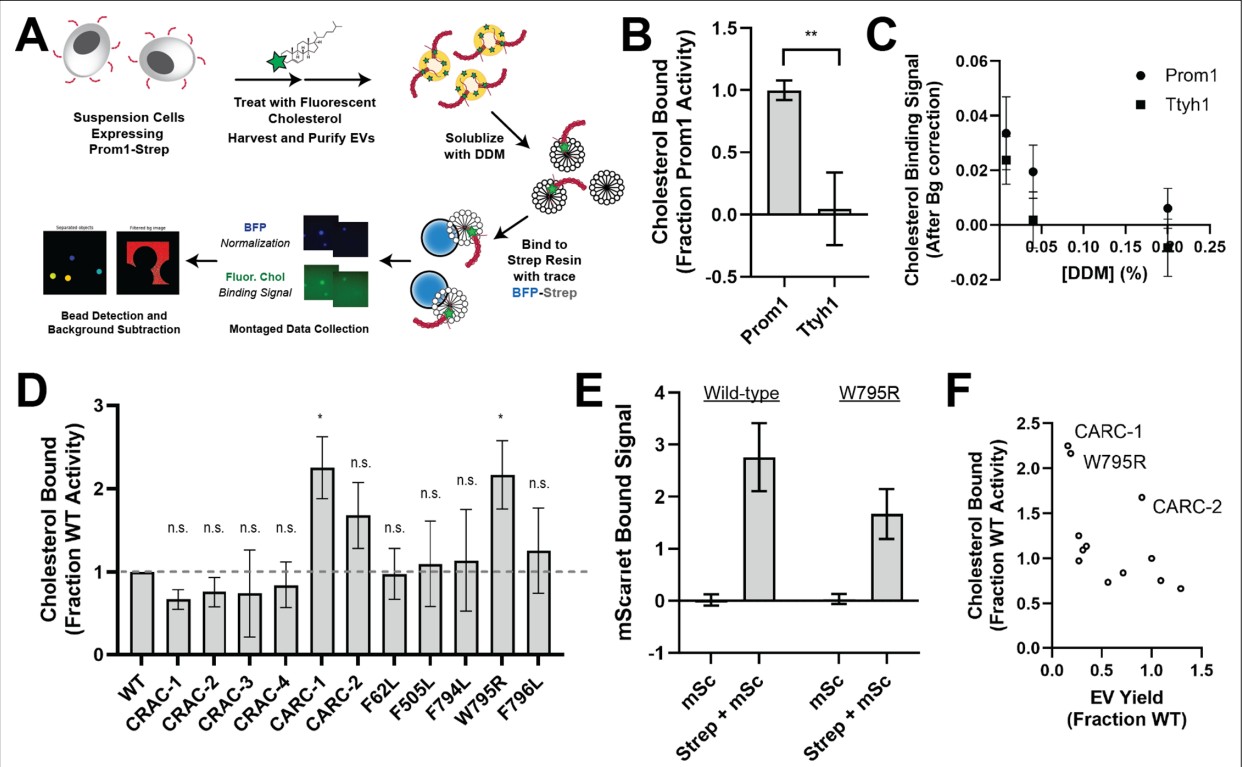

**Figure 4.** Ttyh1 binds cholesterol less stably than Prom1. (**A**) Cholesterol co-immunoprecipitation (chol-IP) assay graphic protocol. (**B**) Comparison of relative cholesterol bound by Prom1-Strep or Ttyh1-Strep ($n = 3$, **p < 0.01 by Student's two-tailed unpaired $t$ test). (**C**) Fluorescent cholesterol co-immunopurified by Prom1-Strep or Ttyh1-Strep at different concentrations of $n$-dodecyl-β-D-maltoside (DDM) detergent. Error bars indicate standard deviation (SD) ($n = 3$). (**D**) BODIPY-cholesterol-binding measurements for WT and mutant variants of Prominin 1 (Prom1). Error bars indicate SD ($n = 3$, n.s.p > 0.0045, *p < 0.0045 by Student's two-tailed unpaired $t$ test with significance thresholds adjusted by Bonferroni correction). (**E**) Red fluorescence signal from anti-Strep immunopurification of DDM-solubilized extracellular vesicles (EVs) from cells expressing Prom1-mScarlet (mSc) or both Prom1-mScarlet and Prom1-Strep (Strep + mSc). Error bars indicate SD ($n = 3$). (**F**) Comparison between EV yield (*Figure 2C*) and chol-IP fluorescent cholesterol binding (**D**), with notable outliers labeled.

The online version of this article includes the following figure supplement(s) for figure 4:

**Figure supplement 1.** Fluorescent cholesterol binding by Prom1 and Ttyh1.

chol-IP) to quantify interaction of Prom1 with fluorophore-labeled cholesterol (*Figure 4A*). Prom1-Strep or Ttyh1-Strep were transfected into Expi293 cells, and cells were labeled with a low concentration of fluorescent cholesterol. EVs were purified from the CM, solubilized with nonionic detergent (1% DDM), and immunoprecipitated with Strep resin trace labeled with blue fluorescent protein (mTagBFP2-Strep). We then collected epifluorescence micrographs of the resin particles, computationally segmented and filtered each image, and quantified bound cholesterol using mTagBFP2 as a normalizing control (*Figure 4A*). This approach permits sensitive quantification of bound lipid while efficiently excluding autofluorescent and refractive artifacts.

We performed chol-IP analysis on Prom1 or Ttyh1 EVs labeled with BODIPY-cholesterol. Surprisingly, fluorescent cholesterol robustly co-purified with Prom1 but not with Ttyh1 (*Figure 4B*). To verify that this difference is not a result of nonspecific fluorophore–protein interactions, we replicated the observation using a different fluorescent cholesterol analog (AlexaFluor647-cholesterol) (*Figure 4—figure supplement 1*). We further verified that the cholesterol is specifically bound to Prom1 by quantifying bound cholesterol at different concentrations of solubilizing detergent (*Figure 4C*). Nonspecifically associated fluorescent cholesterol in Ttyh1 samples is delipidated at 0.04–0.2% DDM, 5- to 25-fold below the assay working concentration, suggesting that the assay informs on specific lipid binding that is resistant to delipidation. Although this method cannot inform on transient or kinetically unstable protein–lipid interactions, we infer that Prom1 and Ttyh1 differ in how stably they interact with membrane cholesterol.

Given the striking difference in cholesterol-binding stability between Prom1 and Ttyh1, we next asked whether Prom1 mutants also disrupt cholesterol interaction stability. We purified BODIPY-cholesterol-labeled EVs produced by WT and mutant forms of Prom1 and subjected equal input concentrations of Prom1 to chol-IP analysis. To our surprise, none of the mutants substantially disrupted cholesterol binding, but the W795R mutant and the CARC-1 mutants bound cholesterol significantly more stably than WT Prom1 (*Figure 4D*). We chose to focus our efforts on the W795R mutant as it is a naturally occurring single-residue mutation with exceptional evolutionary conservation and a clinically validated disease phenotype. To verify that differences in cholesterol binding between wild-type and W795R Prom1 do not arise from a gross oligomerization defect, we purified EVs from cells co-transfected with Strep- and mScarlet-tagged Prom1 and measured co-purification of Prom1-mScarlet on Strep resin. Prom1 W795R co-purified with ~60% as much mScarlet fluorescence as WT Prom1, indicating that W795R indeed multimerizes, albeit with reduced efficiency (*Figure 4E*). Comparative analysis of EV Yield (*Figure 2C*) and cholesterol bound (*Figure 4D*) indicates that though there may be a weak inverse correlation between cholesterol binding and stable cholesterol binding, W795R, CARC-1, and CARC-2 are outliers that bind cholesterol exceptionally tightly (*Figure 4F*).

## Cholesterol-binding stability inversely correlates with Prom1 EV membrane bending

Because Ttyh1 promotes greater membrane curvature than Prom1 but does not stably bind cholesterol, we hypothesized that cholesterol-binding stability may contribute to negative regulation of membrane bending by Prominin-family proteins. If this is the case, then W795R Prom1, which binds cholesterol more stably than WT protein, should not produce EVs that exhibit the tubular morphology observed in Ttyh1 EVs. After purifying W795R Prom1 EVs, we observed a similar solution size for WT (164 ± 14 nm) and W795R EVs (186 ± 13 nm) (*Figure 5—figure supplement 1*). We measured the size and shape of W795R EVs by cryo-TEM and observed largely spherical vesicles with some local deformations, similar to what we observed with WT Prom1 EVs (*Figure 5A*). Generally, WT and W795R Prom1 EVs are both spherical, with W795R having no vesicles that fall into our tubular morphology classification (*n* = 1211) (*Figure 5B*). Though both large and small diameter EV populations were observed by cryo-TEM, W795R Prom1 had a larger fraction of small EVs than WT Prom1, indicating possible decreased stability or increased fissile propensity in the W795R Prom1 EVs (*Figure 5C*).

If stable cholesterol binding negatively regulates membrane bending, then depleting cholesterol away from EV membranes could induce a Ttyh1-like morphology in Prom1 EVs. We tested this model directly by purifying WT Prom1 or Ttyh1 EVs and treating them with methyl-beta cyclodextrin (mBCD), a compound that extracts cholesterol from membranes (*Röper et al., 2000*; *Zidovetzki and Levitan, 2007*). After re-purifying EV samples away from free mBCD and mBCD–cholesterol complexes, Prom1 EVs treated with mBCD (2.5–10 mM) exhibited a decrease in cholesterol content (*Figure 5D*). A control set of Ttyh1 EVs treated with 0 or 10 mM mBCD showed a similar change in cholesterol content (*Figure 5D*). Treated and untreated EVs were then analyzed by NS-TEM to compare EV size and morphology (*Figure 5E*). We observed a 5.4-fold increase in the fraction of Prom1 EVs that deviated from spherical membrane topology after 10 mM mBCD treatment compared to untreated EVs (3.0% vs 0.6% of EVs, *n* = 232 and *n* = 356, respectively) (*Figure 5F*). Importantly, identical treatment of Ttyh1 EVs was used as a nonspecific control for cholesterol depletion from EVs. Ttyh1 EVs treated with mBCD under identical conditions only showed a 1.4-fold increase in deformed EVs (2.1% at 10 mM mBCD vs 1.5% in mock-treated EVs, *n* = 433 and *n* = 535, respectively) (*Figure 5F*). Thus, cholesterol-depleted Prom1 EVs are more prone to membrane deformation than untreated EVs.

## Discussion

### Prom1 and Ttyh1 bend membranes and form EVs

We report here that both Prom1 and Ttyh1 can induce EV formation in cultured cells upon overexpression (*Figures 1B and 3C*). Though we have not examined whether Ttyh1 endogenously induces EV formation, we note that it is sufficient to do so in a recombinant system with similar or higher efficiency than Prom1. Given similar membrane remodeling behavior of Prom1 and Ttyh1 in cell culture (*Hori et al., 2019*; *Stefaniuk et al., 2010*), Ttyh1 EVs may form by a similar mechanism to Prom1 EV

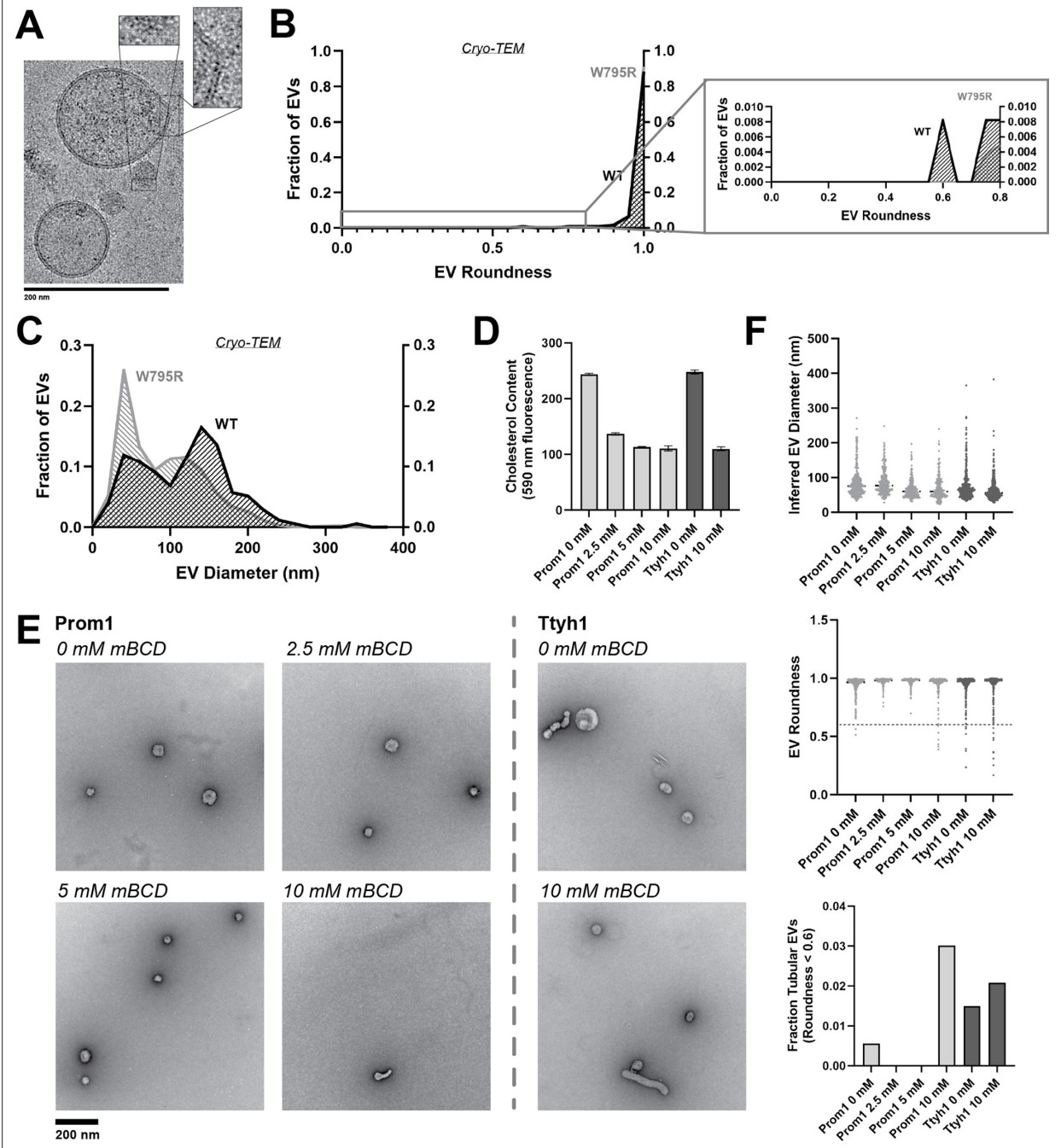

**Figure 5.** Cholesterol binding stability inversely correlates with Prom1 EV membrane bending. (**A**) Representative cryo-transmission electron microscopy (cryo-TEM) images of W795R Prom1-Strep extracellular vesicles (EVs). Magnified insets show bilayer density. Images are lowpass filtered to 5 Å to enhance contrast. (**B**) Comparison of WT or W795R EV roundness in cryo-TEM images. Secondary plot only includes EVs with roundness ≤0.8 (n = 176 and n = 1211 for WT and W795R Prominin 1 (Prom1) cryo-TEM measurements, respectively.) (**C**) Comparison of WT or W795R Prom1 EV diameter in cryo-TEM images (n = 122 and n = 821 for WT and W795R Prom1 cryo-TEM measurements, respectively). (**D**) Cholesterol content of Prom1- or Ttyh1-purified EV samples after treatment with methyl-beta cyclodextrin (mBCD). Error bars indicate standard deviation (SD) (n = 3). (**E**) Representative negative-stain transmission electron microscopy (NS-TEM) images of purified mBCD-treated EVs. (**F**) Quantification of EV diameter (*top*) and roundness (*middle*) from NS-TEM images, as well as quantification of tubulated EVs (roundness <0.6) (*bottom*).

The online version of this article includes the following figure supplement(s) for figure 5:

**Figure supplement 1.** Dynamic light scattering distributions for Prom1 WT and W795R variants.

formation. We find that Prom1 and Ttyh1 form EVs that are of similar size and that the two proteins exist in the same pool of EVs when co-expressed (*Figure 3K, L*).

Membrane bending by Prom1 is critical for maintaining outer segment membrane homeostasis in photoreceptors. Intriguingly, photoreceptors with impaired expression of the disc rim stabilizing protein Peripherin accumulate EVs 200–280 nm in diameter at the outer segment base (*Salinas et al., 2017*). We observe formation of Prom1 EVs of relatively similar size in our reconstituted system (*Figure 1—figure supplement 1*). This may be an example of endogenous regulation of Prom1-induced membrane curvature, with curvature-stabilizing proteins like Peripherin preventing the evaginating membrane from budding into EVs.

## Ttyh1 forms EVs with membranes that are more tubulated than Prom1 EV membranes

Ttyh1 produces EVs similar in size and density to Prom1 EVs (*Figure 3—figure supplement 1*, *Figure 3J–L*). However, Ttyh1 EVs are dramatically more tubulated that Prom1 EVs (*Figures 1E, F and 3D, E, G*). Several factors could contribute to this increase in membrane bending. First, Ttyh1 is present in EVs at concentrations approximately fivefold higher than Prom1 (*Figure 3H, I*). Membrane proteins can induce curvature through molecular crowding when enriched at sufficiently high local concentrations (*Derganc and Čopič, 2016*). Second, Ttyh1 co-purifies with cholesterol less efficiently than Prom1 does (*Figure 4B*). A confounding effect in our experiments is that molecular crowding may also alter the efficiency with which cholesterol is extracted from EV membranes by detergent. Although we did not observe a kinetically stable interaction between Ttyh1 and cholesterol, it remains to be conclusively determined whether Ttyh1–cholesterol interactions occur in native membranes. Ideally, molecular-crowding and cholesterol-binding effects could be differentiated by modulating protein-to-lipid ratios in EV membranes. Unfortunately, our native purified EVs do not provide a tractable path to perform these experiments. Future experiments reconstituting Prom1 and Ttyh1 into artificial liposomes with variable cholesterol content will be necessary to fully elucidate the mechanistic differences in membrane bending between Prominin and Ttyh proteins.

## Cholesterol-binding stability may contribute to membrane-bending regulation by Prom1

We find here that Prom1 in EVs forms a stable interaction with cholesterol that is resistant to delipidation by nonionic detergent (*Figure 4B, C*). A disease-associated mutation in a strikingly conserved residue in the fifth transmembrane helix of Prom1, W795R, substantially stabilizes cholesterol binding (*Figures 2A and 4D*). In AlphaFold2 models, Trp-795 forms a series of base–stacking interactions with Phe residues in neighboring transmembrane helices (*Figure 2B*; *Jumper et al., 2021*). We find that conservative mutations in several of these adjacent aromatic residues impair EV formation by Prom1, but do not mimic the stable cholesterol binding of W795R (*Figures 2C and 4D, F*). Cholesterol is asymmetrically distributed in eukaryotic membranes, with a bias toward the outer leaflet (*Lorent et al., 2020*). Prom1 Trp-795 is predicted to be positioned toward the outer leaflet, possibly potentiating interactions with cholesterol. We also observe increases in cholesterol-binding stability for mutations to two CARC domains in Prom1 (*Figure 4D, F*). This effect could arise because these residues interface with cholesterol or because they destabilize the transmembrane domain in a manner that mimics the effect of the W795R mutation. Given the poor evolutionary conservation of the two CARC motifs in human Prom1, we favor the latter explanation.

We further show that the W795R mutation prevents efficient trafficking of mStayGold-tagged Prom1 to the plasma membrane (*Figure 2E*). W795R Prom1 is expressed but primarily localizes to WGA[+] intracellular vesicles. This observation explains the low yield of W795R EVs produced in cell culture (*Figure 2C*). Several studies have demonstrated that protein trafficking through the endomembrane system is sensitive to cholesterol levels (*Ridsdale et al., 2006*; *Stüven et al., 2003*), and our findings may suggest an as-yet undefined role for cholesterol in Prom1 membrane trafficking as well.

## Prominin and Ttyh proteins are homologous proteins that both promote EV formation

Prominin and Ttyh proteins are both biologically implicated in membrane bending. Prominins localize to cholesterol-rich domains of the plasma membrane and drive protrusion-shed EV formation in

differentiating stem cells (*Röper et al., 2000*; *Marzesco et al., 2005*; *Dubreuil et al., 2007*). Ttyh proteins are associated with dendritic spikes in healthy neurons and with tumor microtube formation in aggressive astrocytoma (*Matthews et al., 2007*; *Jung et al., 2017*; *Stefaniuk et al., 2010*; *Wiernasz et al., 2014*). Given their sequence homology and transmembrane topology, and their similarities in membrane localization, membrane bending, and EV formation behavior, we suggest that Prom and Ttyh proteins be considered members of a broader prominin family of membrane remodeling proteins. Further work will be required to fully characterize the functional specialization of Prom and Ttyh proteins.

## Materials and methods
### Expression constructs
All Prom1 expression constructs were generated by site-directed mutagenesis from pCS2-Prom1-YFP (a gift from N. Sasai, Nara Institute of Science and Technology). The base Prom1 construct used is the human S1 isoform (NCBI accession NP_001139319.1) (*Fargeas et al., 2007*). Prom1-Strep was subcloned into pLV-EF1a vector (a gift from K. Hochedlinger, Massachusetts General Hospital) for lentiviral transduction. Human Pcdh21 isoform 1 (NCBI accession NP_149091.1) was synthesized (GenScript) and cloned into a pCDNA3.1 vector (Thermo Fisher) for mammalian cell transfection. mTagBFP2 was subcloned from pBAD (Addgene #54572, RRID:Addgene_54572, a gift from Michael Davidson) into a pET28a vector with a C-terminal Strep tag for bacterial overexpression. Ttyh1 was expressed from a pLX304 vector after addition of C-terminal Strep and His tags to an existing construct (Addgene #161676, RRID:Addgene_161676, a gift from Mike McManus). To stably express fluorescently tagged Prom1 WT and W795R variants in HeLa cells for live cell imaging, Prom1-StayGold (RRID:Addgene_210821) and Prom1[W795R]-StayGold (RRID:Addgene_210822) sequences were synthesized (GenScript) and cloned into a P2555 vector (a gift from S. Jakobs, Max Planck Institute for Biophysical Chemistry) to yield constructs pAH18 and pAH20, respectively. Sequences for constructs generated in this study can be accessed at https://www.addgene.org/browse/article/28243505/.

### Cell line construction
Cell lines were regularly tested for mycoplasma contamination. Lentiviral transduction of Prom1-Strep into Expi293 cells was performed using a modification of published protocols (*Elegheert et al., 2018*; *Kutner et al., 2009*). Briefly, lentiviral particles were produced in 293T cells (ATCC #CRL-3216, RRID:CVCL_0063) by transient transfection of pLV-EF1a-Prom1-Strep with VSVG (RRID:Addgene_98286) and Delta 8.9 plasmids (a gift from K. Hochedlinger, Massachusetts General Hospital) using the Lipofectamine 3000 system (Thermo Fisher) and incubated overnight at 37°C, 5% $CO_2$. Expi293 cells (Thermo Fisher #A14528, RRID:CVCL_D615) raised for several passages in suspension culture were seeded onto adherent tissue culture plates in adherent culture media (Dulbecco's Modified Eagle Medium (DMEM) (Gibco) supplemented with 10% fetal bovine serum (Gibco) and 1× penicillin/streptomycin (Gibco)) and incubated overnight at 37°C, 5% $CO_2$ to form an adherent monolayer. Transfected cells were exchanged into fresh adherent culture media. Culture media was harvested after an additional 48 hr, filtered through a 0.45-µm vacuum unit, and concentrated from 70 to 0.3 ml in Phosphate Buffered Saline (PBS) buffer by ultracentrifugation according to established protocols (*Kutner et al., 2009*). Adherent Expi293 cells were infected with concentrated lentiviral particles at 75% confluency in adherent culture media supplemented with 8 µg/ml polybrene (Sigma-Aldrich), then incubated for 48 hr at 37°C, 5% $CO_2$. Cells were gently washed with PBS, exchanged into adherent culture media, and incubated for 24 hr at 37°C, 5% $CO_2$. Subsequently, the cells were subjected to antibiotic selection by exchange into adherent culture media containing 2 µg/ml blasticidin (Gibco) for 10 days with regular exchange into fresh selective media and passaging to prevent cells from achieving full confluency. Selection was considered complete when the majority of cells died and antibiotic-resistant foci recolonized the culture plate. The cells were then trypsinized (Gibco) and transitioned back to suspension culture in modified suspension culture media (Expi293 media (Gibco) supplemented with 1% fetal bovine serum) at a density of $1.0 \times 10^6$ live cells per ml of culture and incubated for 48 hr at 37°C, 8% $CO_2$ with 125 rpm rotation. Once the suspension culture reached a density of $3.0 \times 10^6$ live cells per ml of culture, cells were re-passaged in $1.0 \times 10^6$ live cells per ml

of culture in fresh modified suspension culture media supplemented with 1.5 µg/ml blasticidin, and subsequently re-passaged into this media condition every 2 days.

Stable cell lines were constructed using lentiviral and Adeno-associated virus (AAV) transduction methods according to established protocols (*Elegheert et al., 2018*; *Kutner et al., 2009*). For generation of stable HeLa cells expressing WT Prom1-StayGold and Prom1[W795R]-StayGold off the AAVS1 locus, the donor plasmids pAH18 or pAH20 were co-transfected with the nuclease plasmid PX458-AAVS1 (a gift from S. Jakobs, Max Planck Institute for Biophysical Chemistry, RRID:Addgene_113194) using Lipofectamine 3000 (Thermo Fisher). Transfected cells were selected with 10 µg/ml blasticidin (Gibco) starting 48 hr post-transfection for 7 days. Stable clones were expanded for 10 days, and single-cell GRP-positive clones were obtained using a FACS AriaII Cell Sorter (BD Biosciences). After clonal expansion, positive clones were detected and verified by fluorescence imaging.

### Prom1 and Ttyh1 EV reconstitution

Prom1 and Ttyh1 EVs were reconstituted by expression in Expi293 suspension cells. Briefly, Expi293 cells grown in serum-free Expi293 media at 37°C, 8% $CO_2$ with 125 rpm rotation to a density of 3.0 × $10^6$ live cells per ml of culture were transiently transfected with an appropriate plasmid at 1 µg of DNA per 1 ml of culture using the Expifectamine transfection kit (Thermo Fisher) according to the manufacturer's protocol. After 48 hr, cultures were centrifuged for 5 min at 500 × $g$, the media discarded, and the cells resuspended in the same volume of fresh Expi293 media and returned to incubate for an additional 48 hr. After this final incubation, cultures were centrifuged for 5 min at 1500 × $g$ and the CM transferred to clean 50 ml conical tubes.

EVs labeled with fluorescent cholesterol analogs were generated as described above with the following modifications. Two days after transfection, cells were transferred to 50 ml conical tubes, centrifuged for 5 min at 500 × $g$, then resuspended in an equal volume of Expi293 media with fluorescent cholesterol added to a final concentration of 4 µM. Cells were transferred back to shaker flasks and incubated for 24 hr before CM was harvested.

### EV purification

CM was clarified immediately after harvest by centrifuging for 30 min, 10,000 × $g$ at 4°C; then transferring the supernatant to clean tubes and centrifuging again for 1 hr, 21,100 × $g$ at 4°C. The supernatant was transferred to clean 50 ml conical tubes and stored at 4°C until ready for further purification. Clarified CM was transferred to Seton 7030 tubes and each tube underlaid with a 100-µl cushion of 50% sucrose. Tubes were centrifuged in an SW-41 Ti rotor (Beckman Coulter) for 1 hr, 36,000 rpm at 4°C, then ≥200 µl of volume was harvested from the bottom of each tube. In cases where the total harvested volume exceeded 500 µl, the harvested volume was diluted to 11 ml with sterile-filtered PBS buffer, transferred to a final Seton 7030 tube, underlaid with a 100-µl cushion of 50% sucrose, re-centrifuged as described above, and 500 µl of volume harvested from the bottom of the tube. Concentrated EVs were then purified by SEC into sterile-filtered PBS buffer using qEV2-35 nm gravity columns (Izon) at ambient temperature (0.5 ml load volume, 2.5 ml void volume, 1.2 ml harvest volume). Purified EVs were stored at 4°C for up to 8 weeks, over which time no evidence of sample degradation was observed.

### Immunoblots

Protein samples were run on 4–20% or 7.5% Mini-PROTEAN TGX SDS–PAGE gels (Bio-Rad), then transferred to Polyvinylidene fluoride (PVDF) membranes using the TurboBlot semi-dry transfer system (Bio-Rad). Blots were washed briefly three times with 10 ml of PBS with 0.1% (v/v) Tween-20 (PBST) buffer, then incubated for 1–2 hr at room temperature in PBST with blocking agent (5 mg/ml bovine serum albumin (Sigma-Aldrich) for anti-Strep blots, 5% nonfat dry milk for all other blots). Blocking solution was removed and primary antibody solution in PBST with blocking agent (1:2000 rabbit anti-Strep (Abcam #76949, RRID:AB_1524455), 1:4000 mouse anti-Flag (Millipore Sigma #F3165, RRID:AB_259529), 1:2500 mouse AC133-1 (Miltenyi #130-111-756, RRID:AB_2751055)) for 2 hr at ambient temperature or for 12–72 hr at 4°C. The blots were then washed three times for 5–10 min with 10 ml PBST. Secondary antibody solution (1:5000 ECL anti-Mouse (Cytiva #NXA931, RRID:AB_772209) or 1:10,000 ECL anti-Rabbit (Cytiva #NA934, RRID:AB_772206)) in PBST with appropriate blocking agent was then added to the blots and incubated for 1 hr at ambient temperature.

Blots were incubated with 5 ml Western Lighting ECL solution (PerkinElmer) for 1 min and imaged using the chemiluminescence setting on an Amersham 680 gel imager (GE Healthcare). Blots were adjusted for brightness and contrast using GIMP (GNU Project) and annotated with Illustrator (Adobe). After blotting and imaging, images were adjusted for brightness and contrast and subjected to digital densitometry with ImageJ (*Schneider et al., 2012*). Resulting measurements were reported normalized to WT Prom1 on each blot to allow comparisons between blots.

### Dynamic light scattering
DLS measurements were performed using an SZ-100 Nano Particle Analyzer (Horiba). EVs diluted in PBS to a volume of 1 ml were transferred to a disposable plastic cuvette (Fisher) and measurements were taken using settings for polydisperse liposomes in aqueous buffer. All measurements were taken at 25°C in multiple technical replicates to control for instrument sampling error.

### Silver stain
EV samples were run on 7.5% SDS–PAGE Tris–Glycine gels and stained with Pierce Silver Stain for Mass Spectrometry (Thermo Scientific) according to the manufacturer's protocol.

### Glycosylation assays
PNGase F (New England Biolabs) was used to remove *N*-glycan moieties from proteins under denaturing conditions according to the manufacturer's instructions.

### Cryo-TEM sample preparation and imaging
Prom1 and Ttyh1 EVs were vitrified on 300-mesh gold Quantifoil R 1.2/1.3 + 2 nm Carbon grids (Electron Microscopy Sciences). Briefly, grids were glow discharged in an EasiGlow device (Pelco) set to 5 mA, 30 s, 0.39 mbar, with a 15 s vacuum hold time. The grids were then treated with 5 µl of purified EVs, incubated for 60 s to allow EVs to adhere to the carbon layer, then blotted with a VitroBot Mark IV (Thermo Scientific) set to 22°C, 5 s blot time, +15 blot force, 100% humidity; and plunge frozen in liquid ethane. Vitrified samples were imaged on a Titan Krios microscope (Thermo Scientific) with a Falcon 4 direct electron detector (Thermo Scientific) using counted detection mode, 105,000 × nominal magnification, 0.83 Å pixel size, with 49-frame fractionated collection, 49.8 e⁻/Å (*Satir and Christensen, 2007*) total dose, and defocus ranging from −0.8 to −2.0 µm in 0.1 µm increments. Images were processed and analyzed with CryoSparc v. 4.2.1 (Structura Biotechnology). Vesicles were defined and diameter (all EVs) and roundness (only EVs completely visible on one micrograph) were calculated using custom scripts that extend CryoSparc, made publicly available at GitHub, (copy archived at *Bell, 2023*).

### NS-TEM sample preparation and imaging
Formvar carbon film 400-mesh copper grids (Electron Microscopy Sciences) were glow discharged in an EasiGlow device (Pelco) set to 30 mA, 30 s, 0.39 mbar, with a 15 s vacuum hold time. 5 µl of EV sample was applied to a glow-discharged grid and incubated for 60 s at room temperature. The grid was then blotted manually with filter paper (Whatman #43), briefly washed three times with 20 µl of PBS buffer, blotted, washed two times with deionized water, blotted, washed one time with 1.25% (wt/vol) uranyl formate (Electron Microscopy Sciences), and blotted. The grid was then floated for 10 s on a 20 µl of drop of 1.25% uranyl formate, blotted, and allowed to air dry. Imaging was performed on a Tecnai T12 transmission electron microscope (FEI) equipped with an XR16 detector (AMT) operated at an accelerating voltage of 120 kV, 30,000 × nominal magnification, 4.32 Å pixel size, and −1.5 µm defocus. Pixel size was manually calibrated prior to image acquisition using a dedicated calibration waffle grid (Ted Pella). Data collected for EV membrane area quantification were automated using SerialEM (*Mastronarde, 2005*). Vesicles were defined and analyzed in CryoSparc (Structura Biotechnology) as described above for cryo-TEM data.

### Quantification of EV membrane area from NS-TEM images
NS-TEM images were collected using an automated acquisition script. The images were processed to resolve membrane features from background using a custom image processing script made publicly

available at GitHub, (copy archived at *Bell, 2024a*). EV membrane area was calculated for each micrograph as the fraction of pixels in the imaging area that was occupied by membrane features.

## Equilibrium sucrose gradient sedimentation analysis

Linear sucrose gradients were prepared in Seton 7030 tubes using sterile-filtered 5% and 30% (wt/vol) sucrose dissolved in PBS buffer using a Gradient Station IP (BioComp). Freshly prepared gradients were loaded into an SW-41 Ti swinging bucket rotor and 200 µl of the purified EVs were layered atop gradients immediately prior to centrifugation. Samples were centrifuged for 5 hr, 22,000 rpm, 4°C, then fractionated into 13 fractions of 930 µl using the Gradient Station IP. Gradient fractions were analyzed by SDS–PAGE and immunoblotting, then quantified as described above.

## Evolutionary analysis of Prom and Ttyh proteins

We identified sequences homologous to human Prom1 using BLAST (*Sayers et al., 2022*) (RRID:SCR_004870) and InterPro (*Paysan-Lafosse et al., 2023*) (RRID:SCR_006695). Putative prominin sequences were curated to include only sequences with five predicted transmembrane helices by TMHMM (*Sonnhammer et al., 1998*) (RRID:SCR_014935) in a 2 + 2 + 1 pattern, containing large extracellular loops (>300 amino acids) and two small intracellular loops (<25 amino acids). This broad search revealed prominin sequences across the eukaryotic tree ranging from metazoans to green plants. Multiple-sequence alignment was performed using MAFFT (*Katoh et al., 2002*) (RRID:SCR_011811) and phylogenetic relationships inferred using IQ-TREE (RRID:SCR_017254) with MODELFIND for evolutionary model selection (*Minh et al., 2020*; *Kalyaanamoorthy et al., 2017*). Branch supports were calculated using the approximate likelihood ratio test (*Anisimova and Gascuel, 2006*). A smaller tree was also constructed spanning only metazoan sequences, with fungi included as an outgroup for rooting. For direct comparison between Prom and Ttyh sequences, homologs of human Ttyh1 were identified for all species included in the prominin metazoan tree and aligned as described above. Figures showing multiple-sequence alignments were generated using JalView (*Clamp et al., 2004*) (RRID:SCR_006459). Figures showing trees were generated using IcyTree (*Vaughan, 2017*).

Conservation of each residue within the Prom and Ttyh metazoan trees was calculated using the Livingstone and Barton algorithm implemented in JalView (*Livingstone and Barton, 1993*). Conservation scores were then plotted onto the AlphaFold2 structure model (*Jumper et al., 2021*) (RRID:SCR_025454) of human Prom1 or a subunit of the solved cryo-EM structure of human Ttyh1 (*Sukalskaia et al., 2021*) using PyMol (Schrödinger, RRID:SCR_000305).

## Live cell fluorescence microscopy and analysis

Confluent monoclonal HeLa cells stably expressing WT Prom1-StayGold and Prom1[W795R]-StayGold were harvested, seeded onto 35 mm glass-bottom dishes (MatTek Life Sciences) coated with poly-D-lysine (0.1 mg/ml) and allowed to grow overnight at 37°C under 5% $CO_2$. Cells were stained with WGA conjugated with AlexaFluor 647 (W32466, Thermo Fisher Scientific) at 5 µg/ml for 10 min at 37°C, then washed twice with 1× PBS. Cells were placed in Live Cell Imaging Solution (Invitrogen) prior to imaging using a Nikon A1R HD25 point scanning confocal microscope with GaAsP and PMT detectors, equipped with an Apo TIRF 60×/1.49 NA objective lens and Ti2 Z-drive. Temperature, humidity, and $CO_2$ concentrations were controlled with a Live Cell environmental chamber (Oko-Lab). Image acquisition was performed in NIS-Elements (Nikon Instruments Inc, RRID:SCR_014329) and line-scan analysis was performed using Fiji (*Schindelin et al., 2012*) (RRID:SCR_002285).

## Preparation of fluorescent cholesterol analogs

BODIPY-cholesterol was procured commercially (TopFluor cholesterol, Avanti Polar Lipids) and resuspended at 1 mM in ethanol. AlexaFluor647-cholesterol was synthesized from Alkyne Cholesterol and AZDye 647 Azide Plus (Click Chemistry Tools) by Copper(I)-catalyzed azide–alkyne cycloaddition. Briefly, 1 mM AZDye 647 Azide Plus and 2 mM alkyne cholesterol (each from a 10 mM stock prepared in anhydrous Dimethyl sulfoxide) were combined with 2 mM tetrakis(acetonitrile)copper(I) tetrafluoroborate (from a 40 mM stock prepared in ethanol) in a 350 µl reaction brought up to volume with ethanol. The reaction was incubated at 42°C for 30 min then at 70°C for 2 hr with the reaction vessel open to allow solvent to evaporate. Cholesterol was extracted from the final mixture by diluting the solution to 650 µl with PBS buffer and adding 1 ml methanol and 0.5 ml chloroform. The mixture was

centrifuged for 2 min at 14,000 × $g$ and the supernatant transferred to a clean vessel, after which 1 ml of chloroform and 2 ml of glacial acetic acid were added and mixed by vortexing. The solution was then concentrated by evaporation in a SPD1010 SpeedVac instrument (Savant) until dried, then resuspended in 100 µl ethanol. This preparation was considered to be at 5 mM labeled cholesterol (assuming 100% yield) for downstream calculations.

## Protein expression and purification

A plasmid encoding mTagBFP2-Strep was transformed into *E. coli* BL21 (DE3) pLysS, grown at 37°C in LB media to $OD_{600}$ 0.5–0.7, induced with 0.5 mM Isopropyl β-D-1-thiogalactopyranoside (Gold Biotechnology), and harvested after 3 hr of expression. Cells were resuspended in 25 ml per liter of culture of Buffer A (25 mM 4-(2-hydroxyethyl)-1-piperazineethanesulfonic acid (HEPES) NaOH pH 7.5, 500 mM NaCl, 20 mM imidazole, 0.5 mM dithiothreitol (DTT)) supplemented with 10 µM leupeptin (Sigma-Aldrich), 1 µM pepstatin A (Sigma-Aldrich), 1 mM phenylmethylsulfonyl fluoride (Sigma-Aldrich), 1 mg/ml chicken egg lysozyme (Fisher), and 250 U benzonase nuclease (Sigma-Aldrich); and incubated with stirring for 1 hr at 4°C. Cells were then sonicated for 3 min in an ice/water bath with 5 s on/10 s off pulses. The lysate was then clarified by centrifugation in a JA-25.5 rotor (Beckman Coulter) for 45 min, 15,000 rpm, 4°C. The supernatant was then loaded onto two 5 ml HisTrap columns (Cytiva) plumbed in series equilibrated in Buffer A using a peristaltic pump at 1.5 ml/min flow rate. The column was washed with 100 ml of Buffer A and eluted with 20 ml of Buffer B (25 mM HEPES–NaOH pH 7.5, 500 mM NaCl, 300 mM imidazole, 0.5 mM DTT). The protein was found to be ~95% pure by SDS–PAGE, and the concentration of the eluted material was measured using absorbance signal at 280 nm. The sample was divided into small aliquots, flash frozen in liquid nitrogen and stored at −80°C.

## Chol-IP assays

Prior to running chol-IP assays, input EVs were quantified by SDS–PAGE and immunoblotting to ensure equal inputs of Prom1-Strep and/or Ttyh1-Strep in each assay. EVs were mixed 4:1 with buffer CIA (25 mM HEPES–NaOH pH 7.8, 150 mM NaCl, 5 mM $CaCl_2$, 5% DDM (Anatrace)) and incubated for 1 hr at 4°C with end-over-end rotation, protected from ambient light. During incubation, 0.025 µl of StrepTactinXT 4Flow resin (IBA) per condition was equilibrated with buffer CIB (25 mM HEPES–NaOH pH 7.8, 150 mM NaCl, 5 mM $CaCl_2$, 1% DDM) in a single pooled reaction. mTagBFP2-Strep was added to the resin at a ratio of 2 fmol mTagBFP2-Strep per 1 µl of resin, and incubated for 15 min at 4°C with end-over-end rotation, protected from ambient light. After one additional wash with buffer CIB, the resin was then divided equally across low-binding 1.5 ml tubes (USA Scientific) so that each condition being tested plus one negative control condition had equal inputs of Blue Fluorescent Protein (BFP) labeled resin. The DDM-treated EVs were then added to the appropriate resin tube and incubated for 1 hr at 4°C with end-over-end rotation, protected from ambient light. Each condition was then washed twice with buffer CIB for 5 min at 4°C with end-over-end rotation, resuspended in 45 µl of buffer CIB, and stored on ice, protected from light. Each condition was sequentially pipetted with a cut pipette tip onto a glass microscope slide (Fisher) and gently covered with an 18 mm × 18 mm no. 1 glass cover slip (Matsunami) layered on carefully to minimize trapped air bubbles. Montaged images of the resin beads were collected using an Axio Observer TIRF microscope (Zeiss) in epifluorescence mode with a Prime 95B camera (Photometrics) running SlideBook v. 6.0.24 software (3i, RRID:SCR_014423). Custom scripts were then used to identify resin beads in each image using 4',6-diamidino-2-phenylindole (DAPI channel, BFP) signal to both identify beads and normalize fluorescence signal in the Fluorescein isothiocyanate (FITC channel, BODIPY-cholesterol) or CY5 (AlexaFluor647-cholesterol) channels. Images of all resin particles were manually reviewed to ensure the exclusion of air bubbles, or other non-resin fluorescent artifacts from downstream analysis. Analysis scripts have been made publicly available at GitHub, (copy archived at *Bell, 2024b*).

## Immunopurification assays

Co-immunopurification assays from EVs containing fluorescently labeled Prom1 or Pcdh21 were performed as described above for the chol-IP assay, using montaged fluorescence imaging for sensitive

and quantitative measurements. Prom1-mScarlet was imaged using CY3 and Pcdh21-mNeonGreen using FITC filter sets.

## Cholesterol depletion with mBCD

mBCD (Sigma-Aldrich) was dissolved in sterile-filtered PBS buffer to a final concentration of 20 mM, and a twofold serial dilution series was prepared. For each condition (0, 2.5, 5, and 10 mM mBCD), purified EVs were mixed 1:1 with the appropriate mBCD dilution and incubated for 2 hr at 37°C. After the reaction was complete, the EVs were immediately re-purified over a qEV2-35 nm column (Izon) as described above. Cholesterol in the purified samples was quantified using Amplex Red reagent (Thermo Fisher) according to the manufacturer's protocol, with fluorescence measurements taken using a SpectraMax M5 plate reader (Molecular Devices). EV morphology was characterized by preparing NS-TEM grids with mBCD-treated samples and analyzing the resulting images as described above.

## Acknowledgements

We thank D Syau (Harvard Medical School) and members of the Chao lab for helpful feedback, and Z Li, S Sterling, R Walsh, M Mayer, and S Rawson from the Harvard Medical School Cryo-EM Center for support with cryo-TEM data collection. We are grateful to Dr. Richard Bouley at the Microscopy Core of the Program in Membrane Biology (PMB) at Massachusetts General Hospital for providing helpful advice and equipment access for live-cell confocal microscopy. We thank the staff at the Center for Regenerative Medicine (CRM) Flow Cytometry Core Facility for helpful support in cytometry sorting and analysis. This work was supported by National Institutes of Health R35GM142553 (to LHC) and the Helen Hay Whitney Foundation (to TAB). The content is solely the responsibility of the authors and does not necessarily represent the official views of the National Institutes of Health.

## Additional information

### Funding

| Funder | Grant reference number | Author |
|---|---|---|
| Helen Hay Whitney Foundation | | Tristan A Bell |
| National Institute of General Medical Sciences | R35GM142553 | Luke H Chao |

The funders had no role in study design, data collection, and interpretation, or the decision to submit the work for publication.

### Author contributions

Tristan A Bell, Conceptualization, Resources, Data curation, Software, Formal analysis, Supervision, Funding acquisition, Validation, Investigation, Visualization, Methodology, Writing – original draft, Project administration, Writing – review and editing; Bridget E Luce, Conceptualization, Resources, Data curation, Formal analysis, Validation, Investigation, Visualization, Methodology, Writing – original draft, Writing – review and editing; Pusparanee Hakim, Conceptualization, Resources, Data curation, Formal analysis, Supervision, Validation, Investigation, Visualization, Methodology, Writing – original draft, Writing – review and editing; Virly Y Ananda, Resources, Data curation, Software, Formal analysis, Validation, Investigation, Visualization, Methodology, Writing – review and editing; Hiba Dardari, Data curation, Formal analysis, Supervision, Validation, Investigation, Visualization, Methodology, Writing – review and editing; Tran H Nguyen, Resources, Data curation, Formal analysis, Supervision, Validation, Investigation, Visualization, Methodology, Writing – review and editing; Arezu Monshizadeh, Resources, Data curation, Validation, Investigation, Visualization, Methodology, Writing – review and editing; Luke H Chao, Conceptualization, Resources, Supervision, Funding acquisition, Project administration, Writing – review and editing

## Author ORCIDs

Tristan A Bell http://orcid.org/0000-0002-3668-8412
Pusparanee Hakim https://orcid.org/0000-0002-9018-8179
Hiba Dardari https://orcid.org/0009-0008-8794-6697
Arezu Monshizadeh https://orcid.org/0009-0005-4326-7754
Luke H Chao https://orcid.org/0000-0002-4849-4148

## Decision letter and Author response

Decision letter https://doi.org/10.7554/eLife.100061.sa1
Author response https://doi.org/10.7554/eLife.100061.sa2

---

## Additional files

### Supplementary files

• MDAR checklist
• Supplementary file 1. Prom1 mutants used in this study.

### Data availability

Electron microscopy datasets and all biochemical data are deposited in a Zenodo repository. Custom software packages are available at GitHub under a GNU General Public License v3.0: vesicle-quant, vesicle-quantification, and bead-assay. Materials generated in this study will be made available on request.

The following dataset was generated:

| Author(s) | Year | Dataset title | Dataset URL | Database and Identifier |
|---|---|---|---|---|
| Bell TA | 2023 | Electron Microscopy Data for Quantification of Extracellular Vesicles | https://zenodo.org/records/10034616 | Zenodo, 10.5281/zenodo.10034615 |

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
