## [Editor Report]

This work is important because it significantly advances our understanding of membrane protein functions by establishing the mechanisms by which Prominin-1 and Tweety Homology-1 induce extracellular vesicle (EV) formation. The evidence supporting these findings is compelling as it features rigorous biochemical assays and thorough experimental validation. The study will therefore be of particular interest to researchers in cell biology and membrane dynamics, particularly to researchers with a keen interest in EV formation and its implications for cellular processes.

---

## [Decision Letter]

[Editors' note: this paper was reviewed by Review Commons.]

---

## [Author Response]

Reviewer #1 (Evidence, reproducibility and clarity (Required)):SummaryBell et al. overexpress Prom1 or Ttyh1 and test its effect on EV formation from cell lines. They find that Ttyh1 expression leads to an increase in small EVs as well as tubulated EVs, while Prom1 expression leads to a milder increase in small EVs. EV induction by Prom1 is dependent on cholesterol and the authors show that Prom1 makes the cholesterol in EVs more resistant to detergent. The authors show no connection between Ttyh1 EV induction and cholesterol, although they claim it is important. They also show that a disease mutation in Prom1 decreases Prom1 trafficking to the plasma membrane and increases cholesterol resistance to detergent in EVs. The authors also find that the disease mutation decreases the size of the Prom1-induced EVs.Major CommentsResults – line 99-106 – The EV isolation protocol would remove large EVs like the Prom1+ midbody remnants. It is important to explicitly specify that this study focused on small EVs.

We agree with the reviewers and appreciate the suggestion to make this distinction. We have clarified the Results text (lines 104-105) to specify that our method specifically reconstitutes and isolates small EVs.

Statistics – The t tests appear to have been performed without correction for multiple comparisons (Figure 2C-D, Figure 4D). Given that >10 comparisons were made, this can alter the biological significance of p<0.05 (1 incorrect in 20 comparisons). Please reanalyze with a more appropriate statistical test for multiple comparisons (i.e. ANOVA) or apply a correction to the t test values (i.e. Bonferroni).

We agree with the reviewers that multiple test correction is appropriate for these figures. We have applied Bonferroni correction to the t-tests in Figures2C, 2D, and 4D by adjusting our significance thresholds (α), and included additional text in the figure legend to indicate how and why the correction was performed.

The DLS data does not appear to give any insight into EV size (unlike the EM data) and could be removed from the whole manuscript (or moved to supplemental). The authors should also remove any conclusions based on the DLS data.

We appreciate the reviewers raising this point and agree that the DLS is less informative than our other measurements of EV size and morphology. We have moved all DLS figure panels where EV size is characterized by another method to the Supplement.

Discussion – line 382-383 "Because Prom1 EVs arise directly from blebbing of the plasma membrane23, this finding suggests that Prom1 and Ttyh1 traffic to similar regions of the plasma membrane."The authors have not examined where Prom1 or Ttyh1 localize in the plasma membrane and can not draw this conclusion. That both proteins promote plasma membrane budding would only suggest that both proteins localize to the plasma membrane, not subregions of the plasma membrane. However, the authors have not demonstrated that Ttyh1 specifically induces plasma membrane budding. The different size of Ttyh1 EVs could be due to different biogenesis mechanisms (i.e. derived from intracellular organelles instead of the plasma membrane), making this statement an over-interpretation on both parts.

This is a fair point. We have removed this sentence from the Discussion (lines 402-403) as the reviewer requests.

Discussion – line 398-400 "Membrane cholesterol is necessary for Prom1-mediated remodeling20,21 and is present at similar levels in purified Prom1 and Ttyh1 EVs (Figure 5E), indicating that it is undoubtedly important for EV formation by both proteins." & line 415-417 "We find that conservative mutations in several of these adjacent aromatic residues impair EV formation by Prom1, but do not mimic the stable cholesterol binding of W795R (Figures2C, 4D). "The author's data suggests that cholesterol is not important for Ttyh1 to induce EV formation. The authors show that cholesterol depletion does not alter Ttyh1 EV production. Similarly, they find separable effects on cholesterol binding and EV formation with Prom1 mutants, which suggest that there is more to Prom1-mediated EV formation than cholesterol. That cholesterol is present at similar levels can reflect that overexpression of these proteins does not alter the amount of cholesterol in the EV source membrane (i.e. plasma membrane). Also, wouldn't molecular crowding of a membrane protein be predicted to influence how easy it is to extract lipids?

We thank the reviewer for highlighting this imprecisely phrased sentence. We only meant to indicate that cholesterol is present in both sets of EVs and contributes globally to membrane fluidity. We have removed this sentence from the Discussion (lines 419-421) to avoid over-interpretation or confusion.

The reviewer is also correct to point out that molecular crowding could alter how extractable lipids are from EVs. We have included additional explanatory text in the Discussion (lines 421-426) addressing this point.

Discussion – line 431-433 "Our findings suggest that the dynamic interaction of Prom1 with cholesterol may promote efficient maturation and trafficking of Prom1 between the endomembrane system and the plasma membrane.The authors did not investigate whether depleting cholesterol improved Prom1(W795R) trafficking to the plasma membrane, making this inference untested. Soften interpretation or test experimentally.

We appreciate the reviewer raising this point. We have altered the text in this paragraph (lines449-459) to soften our interpretation of these results, as suggested by the reviewer.

Minor CommentsAbstract – "the EVs produced are biophysically similar"The authors don't perform any typical biophysical characterization (beyond size and perhaps density), so do they mean physically similar? Given the Prom1 and Ttyh1 EVs can have different shapes and are significantly different sizes, this statement feels misleading.

We thank the reviewer for pointing out the ambiguity around this word. We agree that “physically similar” is a more precise and accurate term, and have revised all instances of this language in the manuscript.

Intro – line 59-60 – "Large Prom1 EVs (500-700 nm in diameter) appear to form from bulk release of membrane from the cell midbody"Midbody remnants are well defined (if variously named, i.e. flemmingsome) large EVs derived from the spindle midbody, intercellular bridge, and cytokinetic ring. I'm not sure what the authors are trying to express by "bulk release of membrane". Midbody remnants are also a site of membrane tubulation.

The reviewer is correct to point out that midbody remnant release is a well defined process. We originally included this statement to avoid indicating that we are studying the only known class of Prominin EVs, but now recognize that including this creates more confusion that it alleviates. To improve clarity concurrently with the changes referenced above emphasizing that we are specifically studying small EVs, we have removed this reference to the larger class of EVs from the introduction (lines 61-63).

The effect on total numbers of EVs is buried in the y-axes of the EM graphs, making it difficult to distinguish where a higher n of images was examined vs. where there is an increase in EVs. This is especially hard to interpret given the high difference in n values.

The reviewers raise a valid critique of these figure panels. To improve clarity, we have adjusted the y-axes to represent the fraction of EVs rather than the absolute value of EVs, and listed the n values in figure legends.

Figure 2C – Missing WT error bars

We appreciate the reviewer’s concern for the WT error bars in these figures. The measurements underlying these plots are derived from quantification of Western blots. Because the blots have a limited number of lanes, the WT sample was run as a normalization control on each of several sets of blots. By employing this approach, we could make quantitative comparisons within each blot without needing to make direct comparisons between blots, eliminating confounding variables such as blotting times, positions of blots on rotary shakers, developer incubation time, exposure times, etc. Because WT lanes were used for normalization, each “WT” blot condition has its own set of error bars that was used for t-test comparison with the samples that share a blot. For this purely technical reason, we can represent the data either normalized against WT values or with three separate WT measurements for each plot. In the interest of clarity and transparency, we elected to report the values normalized to WT and to include all raw blot images in Supplementary Figure S4. We understand that we could have made this more transparent, so to clarify this decision for readers, we now explicitly reference the raw blot images in both the Results text (lines 185) and in the Figure 2 legend.

Figure 3H, 5C – Why not show raw numbers on the y-axes of the inset graphs like the main graph? Also, if it is only showing a subset of roundness ranges, then the x-axis should not go to 1 (i.e. axis range 0-0.8 would be clearer). I had a hard time figuring out what these insets were trying to show me, so please think about presenting this data more clearly (and larger).

For clarity, we have moved the inset graphs to separate panels alongside the main panel and implemented the requested changes to the axes (see Figures 3G, 5B).

Discussion – line 377 – "Though we do not claim that Ttyh1 endogenously induces EV formation"This statement could be misinterpreted to say that you do not think endogenous Ttyh1 regulates EV formation. Rephrase as "although we have not examined whether…"

We thank the reviewer for pointing out this unclear sentence and have applied the requested change (line 397).

Discussion – line 400-402 "Our results do not indicate that Ttyh1 does not bind cholesterol, merely that it does not form an interaction that is sufficiently kinetically stable to be co-immunoprecipitated."The phrasing here is confusing with multiple "not". It is better to leave things open than to say what you have not shown. Rephrase suggestion: "Although Ttyh1 was not able to form a kinetically stable interaction for co-immunoprecipitation, it remains to be determined whether Ttyh1 is able to bind cholesterol."

We thank the reviewer for their suggestion and have modified the sentence to avoid double-negative phrasing (lines 422-426).

Videos – I'm not sure what the two videos add. It's difficult to convince myself that I see plasma membrane labeling in either video, especially in comparison to the over-exposed WGA staining. Also, why are there ~5 sec of empty movie at the end of each?

We appreciate the reviewer’s feedback and have removed the videos from the manuscript.

Reviewer #1 (Significance (Required)):The data is interesting and well presented, but over interpreted in the discussion.The data on Ttyh1 expression inducing EVs is novel, but limited to overexpression studies.This study will be of interest to the EV, membrane curvature, and Prmn1/Tthy1 fieldsMy expertise is in basic research on membrane trafficking (including EV formation) and lipids

We thank the reviewer for their favorable review and helpful suggestions.

Reviewer #2 (Evidence, reproducibility and clarity (Required)):In this study, authors investigated the role of Prom1 and Ttyh1 proteins on EV formation. They showed that both proteins can induce EV formation, while the mechanisms by which they do it might differ slightly. Ttyh1 binding to cholesterol is not as pronounced as Prom1. Surprisingly, cholesterol binding efficiency inversely correlates with EV formation. Also, EVs induced by Tthy1 and Prom1 are structurally different.My suggestions to improve the manuscript are below.- Figure 2E is not very convincing. As the authors mentioned, the signal is too low to have a concrete conclusion. The line scans somehow show that WT is more membrane-localized than mutant, but colocalization of Prom1 and WGA seems very similar in both cases. Is it certain that the addition of fluorophore did not change the trafficking? Does endogenous Prom-1 staining look like this? Also, why is WGA staining brighter in mutant sample, just a usual variation or biologically important?

We understand the reviewer’s concern about low signal, but respectfully disagree that the signal is too low to draw a meaningful conclusion. The only point we conclusively make in Figure 2E is that WT Prom1 is more efficiently trafficked to the plasma membrane than W795R Prom1. We feel that this effect is sufficiently well evidenced by the line scan analysis in Supp. Figure S5, where Prom1 peaks are cleanly visible for WT but not for W795R protein.

We observe somewhat variable WGA staining in our experiments, and the differences we show in this figure panel are representative of typical staining variation. We do not draw any biological conclusions from the level of WGA present, only from its localization. Because both the plasma membrane and late endosomes are WGA^+^, we suspect that the W795R Prom1 is failing to traffic from endosomes to the plasma membrane. However, given the limitations of our fluorescence assay, we have removed any claim beyond the change plasma membrane trafficking efficiency from discussion of this experiment.

We cannot conclude whether the mStayGold fluorophore alters trafficking of Prom1 to the plasma membrane. In response to the reviewer’s comment, we attempted to use immunofluorescence to measure membrane localization of untagged Prom1 with the AC133-1 antibody. Unfortunately, we were unable to optimize this protocol to achieve sufficient membrane staining for quantification. We have softened our interpretation of Figure 2E in the Results and Discussion (lines 203-204, 450) to acknowledge that the effects we observe are only measured with fluorophore-tagged Prom1.

- I also recommend showing the localization of Ttyh1 on cells.

We appreciate the reviewer’s suggestion here, and it is an experiment we considered. One of the challenges we faced in this assay was quantitatively measuring fluorescent signal along cell-boundary plasma membranes without saturating signal from the very bright WGA^+^ endosomes. Because Ttyh1 globally expresses at higher levels than Prom1 (see Figures 3C, 3I), direct comparison of membrane-localized Prom1 and Ttyh1 is technically challenging in these cells. However, Ttyh membrane localization has been widely reported in other papers (Matthews et al., J. Neurochem, 2007; Jung et al., J. Neurosci., 2017; Sukalskaia et al., Nat. Commun., 2021; Melvin et al., Comm. Biol., 2022) that we now explicitly mention and cite for reader clarity in both the Introduction and Results (lines 69-71, 224-225).

- A graph directly showing cholesterol binding vs EV formation efficiency would be very useful.

We agree with the reviewer that this would be an interesting and useful addition to the paper. We now include this panel in the revised manuscript as Figure 4F.

*-* "Prominin and Tweety homology proteins are homologous and functionally analogous" involves speculation and authors should clearly mention this. Revealing that they are both contributing to EV formation does not make them definitely functionally analogous.

We agree with the reviewer that this sentence is indeed ambiguous and somewhat speculative. We have revised the section heading to “Prominin and Tweety homology proteins are homologous proteins that both promote EV formation” (lines 461-462) to indicate the specific analogous function we observe.

Reviewer #2 (Significance (Required)):Overall, it is a useful addition to the field of cell biology, particularly EV field. EV formation and efficiency are both important topics, and this manuscript might give insights.

We thank the reviewer for their favorable review and helpful suggestions.